# Controlled Structure of Polyester/Hydroxyapatite Microparticles Fabricated via Pickering Emulsion Approach

**DOI:** 10.3390/polym14204309

**Published:** 2022-10-13

**Authors:** Nikita V. Minaev, Svetlana A. Minaeva, Anastasia A. Sherstneva, Tatiana V. Chernenok, Yulia K. Sedova, Ekaterina D. Minaeva, Vladimir I. Yusupov, Tatiana A. Akopova, Peter S. Timashev, Tatiana S. Demina

**Affiliations:** 1Institute of Photon Technologies, Federal Scientific Research Centre “Crystallography and Photonics” RAS, 2 Pionerskaya St., Troitsk, Moscow 108840, Russia; 2Enikolopov Institute of Synthetic Polymeric Materials, Russian Academy of Sciences, 70 Profsoyuznaya St., Moscow 117393, Russia; 3Moscow Aviation Institute (National Research University), 4 Volokolamskoe Shosse, Moscow 125993, Russia; 4Moscow Engineering Physics Institute, National Research Nuclear University MEPhI, 31 Kashirskoe Hwy, Moscow 115409, Russia; 5Institute for Regenerative Medicine, Sechenov First Moscow State Medical University, 8-2 Trubetskaya St., Moscow 119991, Russia; 6Chemistry Department, Lomonosov Moscow State University, Leninskiye Gory 1-3, Moscow 119991, Russia

**Keywords:** microparticles, Pickering emulsion, polylactide, polycaprolactone, hydroxyapatite, nanoparticles

## Abstract

Biodegradable polyester/hydroxyapatite microparticles are widely proposed as microcarriers for drug/cell delivery or scaffolds for bone tissue regeneration. The current research implements the surfactant-free approach for the fabrication of polyester-based microparticles filled with hydroxyapatite nanoparticles (nHA) via the oil/water Pickering emulsion solvent evaporation technique for the first time, to the best of our knowledge. The process of polyester microparticle fabrication using nHA for the oil/water interface stabilization was studied as a function of phase used for nHA addition, which allows the preparation of a range of microparticles either filled with nHA or having it as a shell over the polymeric core. The effect of processing conditions (polymer nature, polymer/nHA ratio, ultrasound treatment) on particles’ total yield, size distribution, surface and volume morphology, and chemical structure was analyzed using SEM, EDX, Raman spectroscopy, and mapping. Addition of nHA either within the aqueous or oil phase allowed the effective stabilization of the oil/water interface without additional molecular surfactants, giving rise to hybrid microparticles in which total yield, size distribution, and surface morphology depended on all studied processing conditions. Preliminary ultrasound treatment of any phase before the emulsification process led to a complex effect but did not affect the homogeneity of nHA distribution within the polymeric core of the hybrid microparticles.

## 1. Introduction

Multicomponent polymer-based microparticles are of great interest due to their research potential and a wide range of applications as catalysts, drug delivery systems, microscaffolds for tissue engineering, building blocks for the fabrication of larger 3D objects, etc. [1,2,3,4,5,6,7]. Diverse functionality of the multicomponent microparticles is explained by a variety of their properties, which could be adjusted as a function of composition and structure [4]. Microparticles comprised of biodegradable polymers (polylactide, polycaprolactone, etc.) are commonly used in drug delivery where slow polymer core degradation enables prolonged drug release [2,4,8,9]. The addition of natural (collagen, chitosan, gelatin, etc.) or inorganic components is often used for producing microparticles tailored as cell microcarriers for tissue engineering so they could mimic the natural matrix for cells to proliferate successfully [10]. The composition of microparticles should correspond to the type of tissue to replace and restore. For example, since bone is a highly organized natural nanocomposite consisting of collagen fibers mineralized with hydroxyapatite nanocrystals [11,12], the application of biodegradable microparticles containing hydroxyapatite (HA) could be beneficial for bone regeneration.

The fabrication method for polymer/HA microparticles affects the particle structure and composition. A variety of methods based on HA deposition onto the pre-formed polymeric microparticle is presented in the literature [13,14]. Hybrid microparticles can be also manufactured using different techniques, such as coacervation, spay-drying, layer-by-layer, sol-gel, etc., but one of the most flexible and usable methods is emulsion solidification [4,15,16,17]. Thus, the oil/water emulsion solvent evaporation technique is based on the emulsification of polymeric solution in the aqueous phase and subsequent evaporation of solvent from the oil phase [18,19]. This method is mainly used to fabricate biodegradable polyester-based particles, which are widely proposed as either drug delivery systems with targeted/prolonged release or as cell microcarriers. In the case of hybrid polyester/HA microparticles, the hydroxyapatite nanoparticles (nHA) could be dispersed in a core polymeric solution which is emulsified in the water phase containing traditional synthetic emulsifiers, such as polyvinyl alcohol or Tween80 [20,21,22,23]. A surfactant-free approach to produce polylactide/HA microparticles could be based on HA synthesis from its precursors during the microparticle formation process [24,25,26]. Another technology for the fabrication of hybrid core/shell microparticles consisting of polyester core and HA shell was proposed by using Pickering emulsion technology [27,28,29,30]. This approach relies on the stabilization of the oil/water interface by nHA dispersed within the aqueous phase, while the oil one consists of polyester solution. Solid nanoparticles of various nature could stabilize the emulsion interface even more effectively than molecular surfactants [31,32,33]. It allows to avoid the usage of additional emulsifiers and to fabricate nHA-armored polyester core/shell microparticles. However, to our knowledge, no works on the fabrication of surfactant-free polyester microparticles filled with nHA have been published. Avoiding the application of molecular surfactants is beneficial in terms of safer usage of microparticles in biomedicine. Such hybrid microparticles are particularly interesting for regenerative medicine both in the form of injectable microscaffolds or as building blocks for the fabrication of larger scaffolds via additive technologies [23,34,35]. A deeper study of the biodegradable hybrid microparticle fabrication process with a controlled structure and morphology is necessary due to rapid progress and an increasing interest in biodegradable particles in the field of tissue engineering.

This work aimed to thoroughly study the surfactant-free fabrication of polyester/nHA microparticles via the oil/water Pickering emulsion solvent evaporation technique, as well as the effects of processing conditions (polymer nature, method of nHA addition, polymer/nHA ratio, ultrasound treatment) on surface and volume structure of the hybrid microparticles. For the first time the surfactant-free approach for the fabrication of polyester-based microparticles filled with hydroxyapatite nanoparticles (nHA) via the oil/water Pickering emulsion solvent evaporation technique was studied.

## 2. Materials and Methods

### 2.1. Materials

Polylactide (PLA, Natureworks, Plymouth, MN, USA) and polycaprolactone (PCL, Polymorfus, Moscow, Russia) were of Mw of 100 kDa. nHA with a size of particles ≤200 nm was purchased from Sigma-Aldrich (CAS: 12167-74-7, Lot #MKBP9910V). All solvents were of analytical grade and used as received from Chimmed (Russia).

### 2.2. Hybrid Microparticle Fabrication via Oil/Water Solvent Evaporation Technique

Microparticles were fabricated via the oil/water solvent evaporation technique, which is based on the liquid-in-liquid emulsion transformation into suspension of solid polymeric microparticles. Two types of hybrid nHA-containing microparticles were fabricated: (1) core/shell microparticles with polymeric core and nHA shell; (2) polymeric microparticles filled with nHA over the whole volume. The main processing difference between core/shell and filled microparticles is the phase into which the nHA were added, water or oil phase, respectively.

The hybrid microparticles with a core/shell structure were fabricated using 6 wt.% solutions of PLA or PCL in CH_2_Cl_2_/acetone mixture (9/1 *v*/*v*) as the oil phase as described in [29]. The aqueous phase consisted of nHA water dispersion with various concentrations of mineral particles (0.1, 0.5, or 1 wt.%). To disperse the nHA in water, we used either magnetic stirring (200 rpm for 10 min) alone or also additionally treated the nHA-containing phase with ultrasound (US) using an immersion ultrasonic dispergator (I-10/0.63, LLC Ultrasonic technique—INLAB, Russia) at 23 kHz for 3 min. The effect of US treatment on the size of nHA aggregates dispersed in water was analyzed using dynamic light scattering (DLS) by Zetatrac (Microtrac, PA, USA). Then, the “oil” polymeric phase was rapidly added by autopipette into the aqueous phase containing nHA nanoparticles under stirring (700 rpm) using a four-bladed propeller stirrer (diameter of 25 mm). The oil to aqueous phase ratio was 1/9 *v*/*v*. The temperature of the emulsion was initially set at 15 °C for the first 15 min of stirring; afterwards, it was increased up to 30 °C to evaporate the solvent from the oil phase for at least 3 h [36]. The formed hybrid microparticles with the polymeric core and mineral shell were washed with mQ water, sieved using apertures ranging from 100 to 400 µm, and freeze-dried. The hollow microparticles that floated at the surface of the aqueous phase at the end of the fabrication process were signed as “capsules”. The weight of each fraction was determined as wt.% of the weight of components dispersed/dissolved within the oil phase. The total yield data are given as a mean value with a standard deviation ranging ±3%. All experiments were carried out at least in triplicate. The scheme of the core/shell microparticle fabrication is shown in Figure 1.

Hybrid microparticles filled with nHA within the volume of polymeric microparticles were fabricated using the nHA dispersion in 6 wt.% solutions of PLA or PCL in the CH_2_Cl_2_/acetone mixture as the oil phase. The amount of the dispersed nHA was varied by the polymer/nanoparticles weight ratios as 1/3, 1/1, or 3/1. nHA were dispersed within the polymeric solution using either a magnetic stirrer (200 rpm, 10 min) only or applying additional US treatment with an aim of ultrasonic bath, using Nordberg NU32D (35 kHz, 3 min) (Spain). The nHA-containing polymeric phase was added within the aqueous phase (mQ) and emulsified at the same conditions as indicated for the core/shell microparticles fabrication. The fabrication process of the polymeric microparticle filled with nHA is shown schematically in Figure 2.

### 2.3. Methods of Microparticle Characterization

The evolution of Pickering emulsions from droplets to microparticles was monitored under optical microscopic observation starting from 5 min after the addition of the oil phase into the aqueous phase, and afterward, every 10 min until the end of the fabrication process (3 h). These aliquots were deposed on glass microscope slides and directly observed in transmission mode using the Livenhuk 320 microscope (Levenhuk, Inc., Tampa, FL, USA).

The formed microparticles were characterized using scanning electron microscopy (SEM) (Phenom ProX, Thermo Fisher Scientific, Waltham, MA, USA) equipped with energy-dispersive X-ray spectroscopy (EDX). Microparticles’ cross-sections were obtained using the ultramicrotome Leica EM UC7 (Leica Microsystems, Wetzlar, Germany) equipped with glass knives. The particles were fixed in epoxy resin (UHU Plus Schnellfest, Baden, Germany) before the slicing.

The distribution of components within microparticles was studied by using a dispersive Raman spectrometer (Nicolet Almega XR, Thermo Fisher Scientific, Waltham, MA, USA). Spectra and spectral maps were registered using an exciting laser with a wavelength of 532 nm. A microscopic 50× (n.a. 0.75) microobjective was used. A 5 µm step was used in the surface mapping mode measuring.

## 3. Results and Discussion

### 3.1. Core/Shell Microparticles Fabricated Using nHA as the Solid Emulsifier in the Aqueous Phase

As a starting point of our research, we evaluated the effectiveness of nHA dispersed in the aqueous phase as stabilizers of the oil/water interface during fabrication of the polyester-based microparticles. Figure 3 represents the effect of concentration, pre-treatment conditions of the nHA water dispersion used as the aqueous phase, and the nature of the polymer dissolved within the oil phase on the total yield, mean size, and size distribution of the fabricated microparticles. As discovered, nHA were well-suited for the stabilization of the oil/water interface without the application of additional molecular emulsifiers. An increase in the nHA concentration within the aqueous phase led to an increase in the microparticle total yield, which could exceed 100 wt.% (total weight of components dissolved within the oil phase, i.e., polyester in this case) due to entrapment of nHA at the phase interface and, therefore, at the surface of microparticles after evaporation of solvents from the oil phase. As can be seen in Figure 3, preliminary US treatment of the aqueous phase gives no significant bonus on microparticle total yield. The US treatment led to a decrease in the content of the microparticles with a size below 100 μm. The fractions of hollow capsules were also lower than that when US was applied. The latter effect could be attributed to the destruction of nHA aggregates to individual nanoparticles, which led to the elimination of air bubbles responsible for air entrapment during microparticle processing [29].

Morphology of the microparticles fabricated under various processing conditions was evaluated using SEM and is presented in Figure 4. Both PLA- and PCL-based microparticles were spherical-shaped and possessed a rather rough surface morphology. However, US pre-treatment of the aqueous phase led to the formation of hybrid microparticles with a deflated-sphere morphology, but only in case of a high nHA concentration in the aqueous phase. At a low nHA concentration (0.1 wt.) in the aqueous phase, the formed PLA-based microparticles were spherical regardless of the use of preliminary US treatment. PCL-based microparticles were spherical only in the case of low nHA concentration without US treatment.

To reveal whether the shape of microspheres (spherical or irregular) is related to emulsification features or the solvent evaporation process, optical microscopy of the emulsion droplets was carried out. Figure 5 shows the kinetics study of the microparticle interface stabilization in time depending on whether the nHA dispersion in the aqueous phase was treated by US or not. Our observations highlighted that at the emulsification onset (5 min after addition of oil phase into aqueous one), the oil droplets were spherical and armored with a layer of nHA. During the solvent evaporation from the oil phase, the polymeric droplets start to lose their spherical shape, but to a different extent as a function of the technique used to disperse nHA within the aqueous phase. In the case of the microparticles fabricated using nHA water dispersion pre-treated with US, the deflated-sphere morphology after solvent removal was observed.

This difference in surface morphology could be related to the possibility of the oil/water interface rearranging upon solvent evaporation. Dynamic light scattering analysis of 1 wt.% nHA water dispersions showed that US treatment reduced the size of nHA aggregates from 1.2 μm to ~100 nm [29]. Thus, in case of usage of nHA after the US treatment, the emulsion interface was stabilized with nHA in the form of mainly individual nanoparticles. The solvent evaporation from the oil phase led to a decrease of the interface surface area, but the rearrangement of the interface was limited due to adsorbed nHA layer. The same morphology of PLA microparticles stabilized with nHA was shown by C.Y. Tham and W.S. Chow [30]. In contrast to their work, we also studied the PLA stabilization with nHA aggregates, i.e., without preliminary US treatment of the aqueous phase. In this case, the oil/water interface could be rearranged by splitting up the nHA aggregates. Figure 6 illustrated the proposed difference in the process of oil/water interface rearrangement. The difference in the shape of PLA and PCL-based microparticles as a function of nHA concentration and US pre-treatment could be explained by different adhesion between nHA and these polyesters [37,38]. The polymer/nHA interaction could affect a variety of processes going on during emulsification and solvent evaporation.

SEM images of the cross-sections showed that these microparticles possessed a core/shell structure (Figure 7a). The chemical structure of the polymeric core was confirmed by Raman spectroscopy. Figure 7b shows Raman spectra of the native PLA polymer and one taken from a core of cross-sectioned PLA-based microparticle.

Energy-dispersive X-ray (EDX) spectroscopy showed a presence of a considerable amount of Ca (7–20 at.%) and P (2–8 at.%) at the microparticle surface, confirming the presence of a shell consisting of nHA (Appendix A). The variations of element concentrations were mostly related to the spots of analysis rather than to microparticle composition and conditions of their fabrication.

Core/shell hybrid microparticles could serve as microscaffolds (i.e., cell microcarriers) for bone tissue regeneration and simultaneous drug delivery. The biocompatibility of similar polylactide-based microparticles was shown previously [39] The inorganic shell may serve for regulating a drug release profile [40,41] and improving cell adhesion to the surface of microparticles [42].

### 3.2. Polymeric Microparticles Filled with Hydroxyapatite Nanoparticles, i.e., the Addition of nHA within the Oil Phase

Hybrid microparticles consisting of the biodegradable polymer filled with nHA over the whole volume are a promising type of injectable carriers for non-invasive drug/cell delivery, but they could be also used as a starting material for fabrication of larger 3D scaffolds for bone tissue engineering via molding or additive technologies, such as laser sintering [34,35,43]. Composite microparticles also appear to be more suitable for such processing than polymeric ones [23]. The distribution of nHA over the whole volume of the polymeric microparticles could provide the constant supply of the mineral bioactive component during the polymeric scaffold degradation. To add nHA as a part of the microparticle core, they were dispersed within the oil phase, i.e., PLA or PCL solution, while deionized water was used as an aqueous phase. We were successfully able to fabricate the microparticles with a total yield ranging from 35 to 72 wt.% (Figure 8). The total yield was logically lower than that of microparticles fabricated using nHA within the aqueous phase since nanoparticles need to migrate from the oil phase volume to the interface first, so the duration of the stabilization process was longer. The mean particle size was logically lower in the case of microparticles made using higher nHA content within the oil phase, while the effect of the oil phase composition on the total yield was more complex and depended on the polymer nature as was also observed in the case of core/shell microparticles. US treatment of the oil phase led to an increase of the total yield of the fabricated microparticles, but mainly by increasing the yield of capsules.

Morphology of the fabricated microparticles depended on the polymer/nHA ratio within the oil phase. As can be seen in Figure 9, the microparticles containing an amount of nHA lower or equal to the polymer had a spherical shape, while the samples fabricated at a polymer/nHA ratio as 1/3 showed the irregular one. The additional US pre-treatment had no significant effect. PCL-based microparticles fabricated at higher nHA content had a disc shape, as was also noticed for the core/shell PCL-microparticles fabricated at higher nHA concentration (1 wt.%) within the aqueous phase (cf. Figure 4 and Figure 9). The difference in microparticle morphology as a function of the polymer nature could be caused by different rates of CH_2_Cl_2_/acetone release from the polymeric oil phase into the aqueous phase during the evaporation process.

EDX spectroscopy of the microparticle surface showed the presence of calcium and phosphorus, confirming the successful nHA migration from the volume of the oil phase to the oil/water interface and their involvement within the surface stabilization process. The concentration of both elements varied significantly as a function of sample and surface point used for probe: Ca in a range of 4–48 at.%; P of 3–14 at.% (Appendix A). Thus, the surface of the polymeric microparticles filled with nHA was enriched with nHA anyway. EDX spectroscopy of various spots at the microparticle cross-sections also confirmed the presence of calcium and phosphorus coming from nHA along with carbon and oxygen elements within the polymeric microparticle volume (Appendix A). However, homogeneity of heteroatom (Ca and P) distribution within the polymeric matrix is not obvious using EDX spectroscopy data.

Mapping of the microparticles’ volume composition was carried out using Raman microscopy. Firstly, Raman spectra of initial components (PLA, PCL, nHA) (Figure 10, spectra 1,2,4) and spots at the cross-section of the microparticles fabricated at 1/1 polymer/nHA ratio (with and without preliminary US treatment) (Figure 10, spectra 3,5) were recorded. Raman spectra showed that the characteristic peak of nHA (960 cm^−1^) does not overlap the main characteristic bands of PLA and PCL, and it is present in the spectra of the cross-sections of the polymeric microparticles fabricated via the nHA addition in the oil phase.

Thus, the peak at 960 cm^−1^ was used for mapping of filler distribution over the volume of PCL- and PLA-based microparticles filled with nHA. The color scale on the spectral Raman maps shows the intensity of the distribution of the characteristic nHA band (960 cm^−1^) within the hybrid microparticles as follows: areas colored in blue correspond to the absence of nHA, green indicates a small amount of nHA, and red indicates the presence of large nHA quantities (Figure 11). Raman maps show that nHA presents within the volume of all the studied samples. The nHA distribution over the cross-section of the microparticles was uniform even without US pre-treatment of the oil phase (nHA dispersion within PLA or PCL solution). However, there are areas with a higher concentration of nHA, which corresponds to the presence of large agglomerates of nHA.

To better evaluate the effect of the preliminary US treatment of the oil phase on the distribution of nHA within the volume of the polymeric microparticles, an additional image analysis was performed. We analyzed several slices of two PCL microparticles made at 1/1 polymer/nHA ratio with or without preliminary US treatment to understand whether the variety within the nHA distribution is related to the processing conditions or to divergence within the slices of the same microparticle. Using the ratio of the intensity of the main spectral line of nHA (960 cm^−1^) to the one of PCL (1110 cm^−1^), spectral maps were made (Figure 12a). The greater the ratio, the more nHA is present in the area. To assess the heterogeneity of the nHA distribution in the polymer core of hybrid microparticles, digital analysis of the resulting spectral maps was performed. At first, these maps were converted to grayscale with a single scale (Figure 12b); then, the normalized histograms of pixel distribution according to their grayness were constructed (Figure 12c). On these histograms, 0 of the horizontal scale corresponds to the pixel in Figure 12b with a black color (no HA), and 1 corresponds to white color (HA content corresponds to the maximum value for all spectral maps).

The histograms in Figure 12c show that the largest part of the cross-section microparticle area (maximum values on the vertical scale) is occupied by points with a nHA content of 25–35% of the maximum value (horizontal scale). This is observed for all investigated microparticles. Thus, the effect of the US pre-treatment of the oil phase on the nHA distribution over the polymeric matrix is insignificant.

## 4. Conclusions

Application of nHA as a solid emulsifier within the aqueous phase allowed the effective stabilization of the oil/water interface, giving rise to core/shell microparticles. The amount of nHA in dispersion affects the shape of the formed microparticles that are spherical only at a low nHA concentration. Ultrasound pre-treatment of the aqueous phase containing nHA had a negative impact on the microparticle formation process since it resulted in deflated-sphere morphology of the hybrid microparticles at high nHA concentration. We also showed, for the first time, the possibility of fabricating polyester microparticles bulk-filled with nHA via the Pickering emulsion technique without application of any molecular emulsifiers. The nanoparticles successfully migrate from the oil phase volume to the interface, initiating the stabilization process. The nHA content within the oil phase affects size and total yield of microparticles. Optimization of the polymer/nHA ratio enables the fabrication of spherical microparticles with a rather high total yield. Interestingly enough, US pre-treatment of either oil or aqueous phase with the aim of promoting nHA disintegration had no real positive effect on total yield and morphology of all the fabricated microparticles. The homogeneity of the nHA distribution was the same for the samples fabricated using either US pre-treatment of the oil phase or nHA mixing in polyester solution via magnetic stirring. However, as confirmed by our results, the difference in polymer/nHA interaction could affect a variety of processes going on during emulsification and solvent evaporation. The obtained hybrid microparticles could serve as microscaffolds (cell microcarriers) for bone tissue regeneration and simultaneous drug delivery. They could also be used as a starting material for the fabrication of larger 3D scaffolds for bone tissue engineering via molding or additive technologies, such as laser sintering.

## Figures and Tables

**Figure 1 polymers-14-04309-f001:**
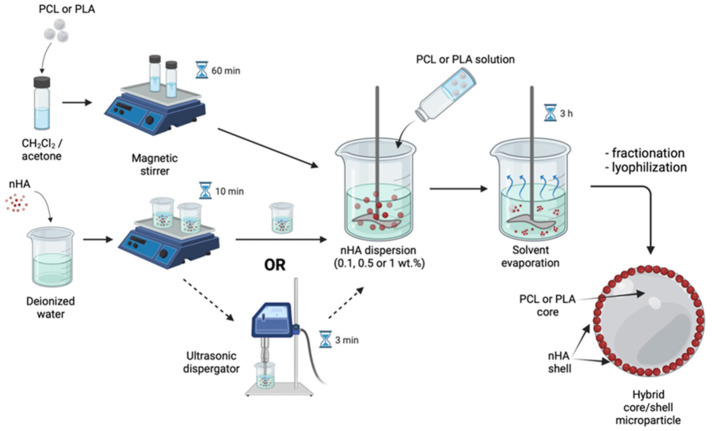
Solvent evaporation technique for core/shell microparticles fabrication. Created with BioRender.com (accessed on 8 September 2022).

**Figure 2 polymers-14-04309-f002:**
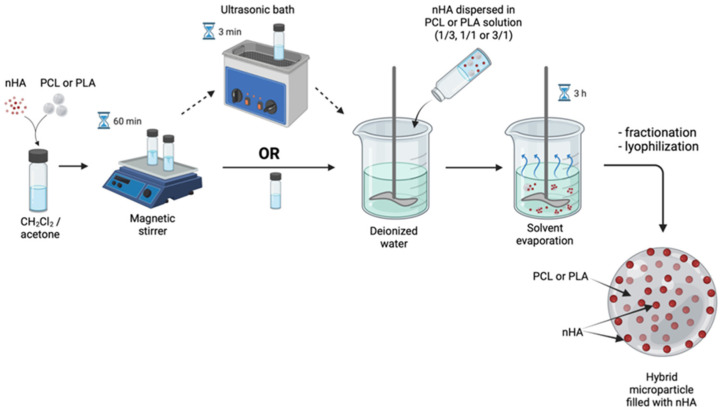
Solvent evaporation technique for the fabrication of filled microparticles. Created with BioRender.com (accessed on 8 September 2022).

**Figure 3 polymers-14-04309-f003:**
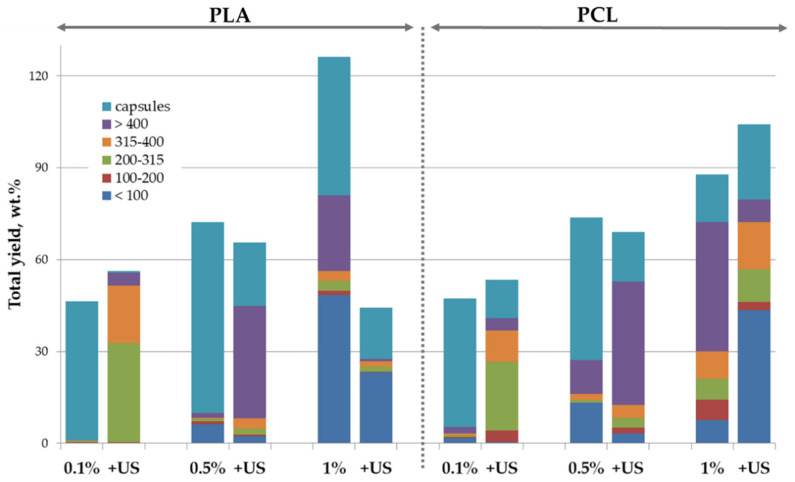
Effect of nature of polymer dissolved within oil phase (PLA or PCL), composition, and pre-treatment conditions of aqueous phase consisted of water dispersion of nHA on the processing yield and size distribution of the hybrid microparticles. Concentration (wt.%) of nHA in aqueous phase is indicated below; US—additional ultrasound treatment of the aqueous phase.

**Figure 4 polymers-14-04309-f004:**
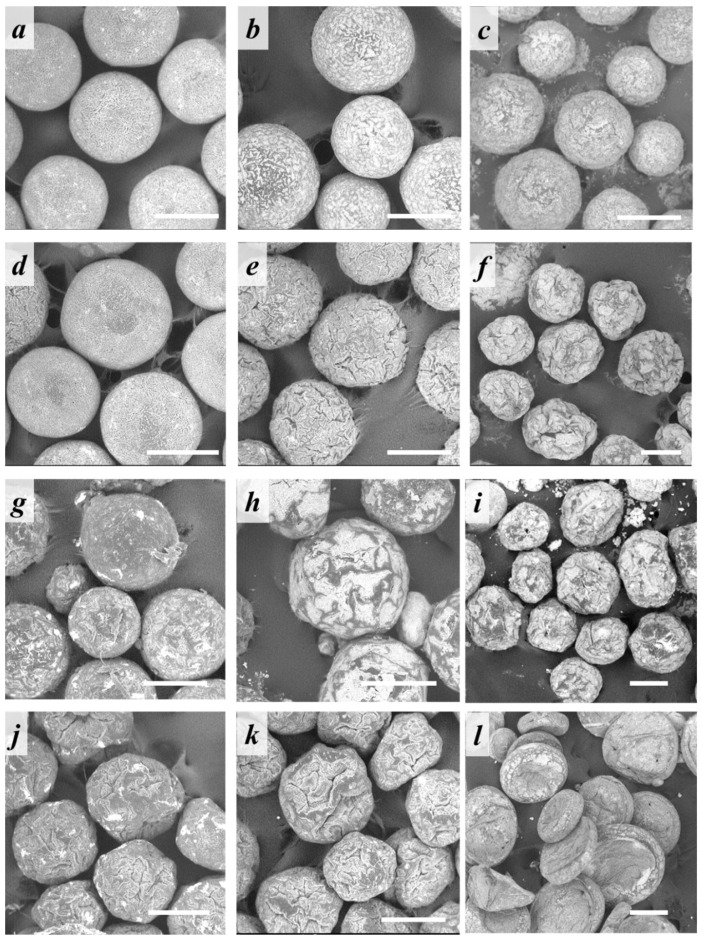
SEM micrographs of core/shell microparticles based on (**a**–**f**) PLA or (**g**–**l**) PCL stabilized with (**a**,**d**,**g**,**j**) 0.1, (**b**,**e**,**h**,**k**) 0.5, or (**c**,**f**,**i**,**l**) 1 wt.% of nHA in the aqueous phase (**a**–**c**,**g**–**i**) without or (**d**–**f**,**j**–**l**) with preliminary US treatment of the water nHA dispersion. Scale bar: 200 µm.

**Figure 5 polymers-14-04309-f005:**
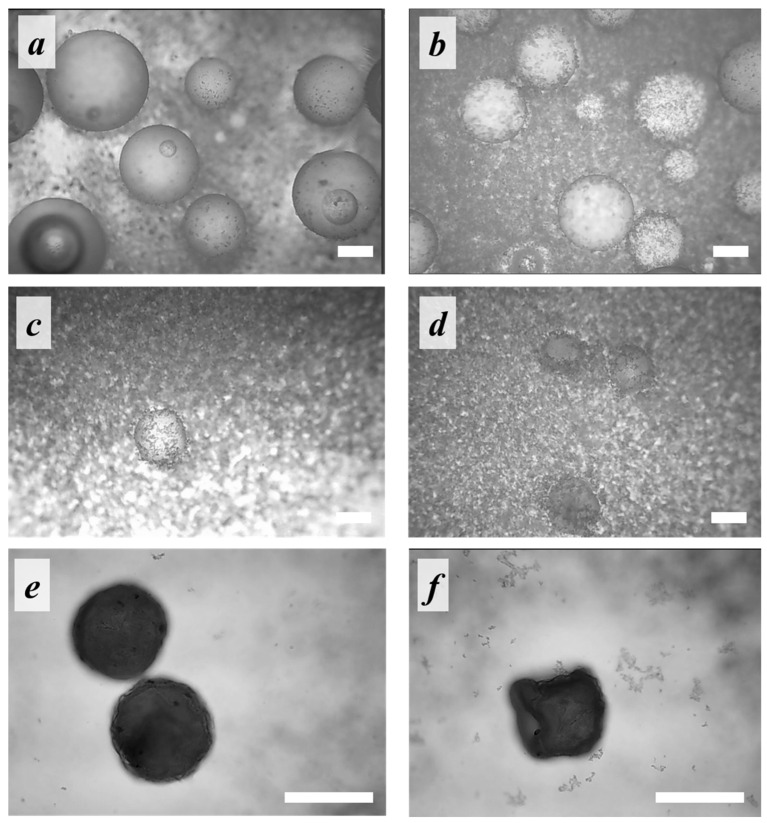
Micrographies acquired under optical transmission microscopes of aliquots of the oil/water emulsion withdrawn at (**a**,**b**) 5 min, (**c**,**d**) 85 min, and (**e**,**f**) 105 min after the dispersion onset. The oil phase—PLA solution; the aqueous phase—1% nHA dispersion obtained using (**a**,**c**,**e**) magnetic stirring and (**b**,**d**,**f**) US-treatment. Scale bar is 100 µm.

**Figure 6 polymers-14-04309-f006:**
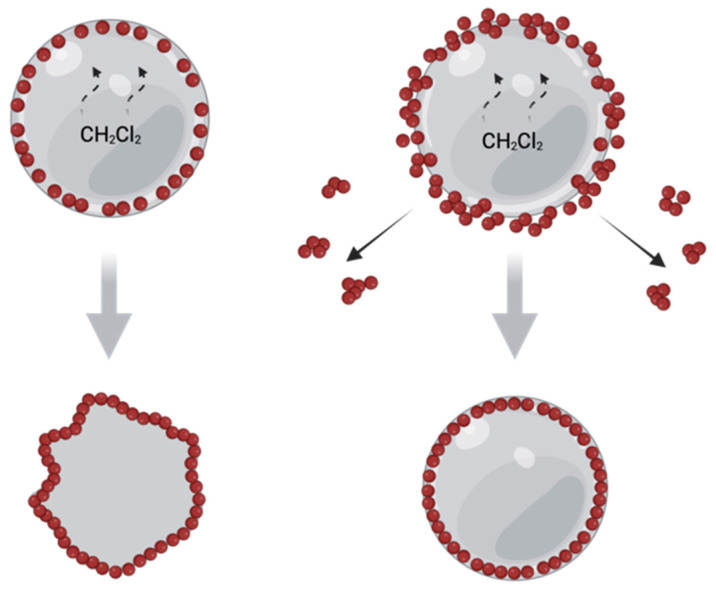
Schematic illustration of supposed reorganization of oi/water interface in the case of droplets stabilized with nHA water dispersions without (**a**) and with (**b**) preliminary US treatment. Created with BioRender.com (accessed on 29 September 2022).

**Figure 7 polymers-14-04309-f007:**
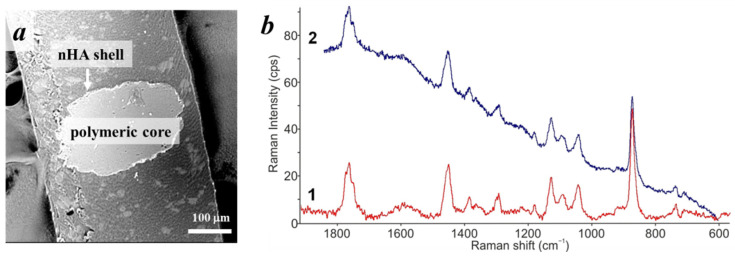
(**a**) SEM image of the cross-section of the PLA-based microparticle stabilized with 1 wt% of nHA without preliminary US treatment of aqueous phase; (**b**) Raman spectra of PLA (1) and from a cross-section of the PLA-based microparticle (2).

**Figure 8 polymers-14-04309-f008:**
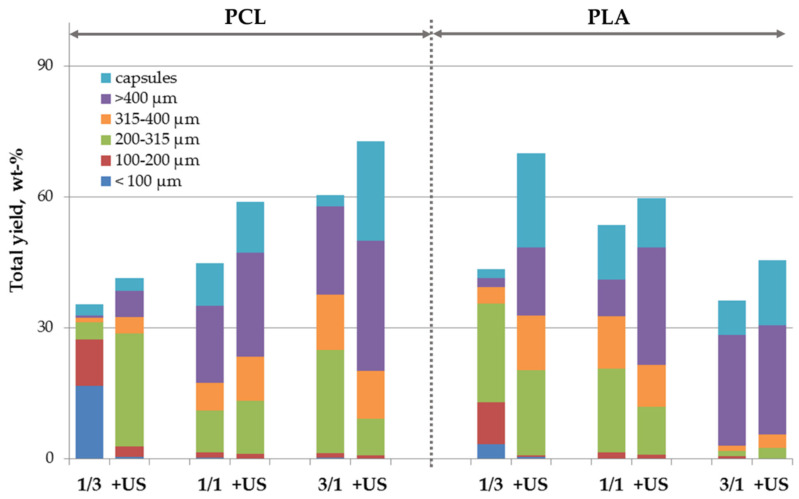
Effect of pre-treatment conditions, composition, and nature of the polymer dissolved within oil phase on the processing yield and size distribution of the polymeric microparticles filled with nHA. Polymer/nHA ratio in the oil phase is indicated below; +US—additional ultrasound treatment of the oil phase.

**Figure 9 polymers-14-04309-f009:**
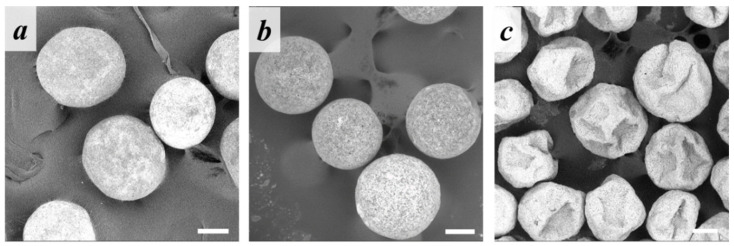
SEM micrographs of the (**a**–**f**) PLA- or (**g**–**l**) PCL-based microparticles containing polymer/nHA ratio in oil phase of (**a**,**d**,**g**,**j**) 3/1, (**b**,**e**,**h**,**k**) 1/1, or (**c**,**f**,**i**,**l**) 1/3 (**a**–**c**,**g**–**i**) without or (**d**–**f**,**j**–**l**) with preliminary US treatment of the oil phase. Scale bars: 100 µm.

**Figure 10 polymers-14-04309-f010:**
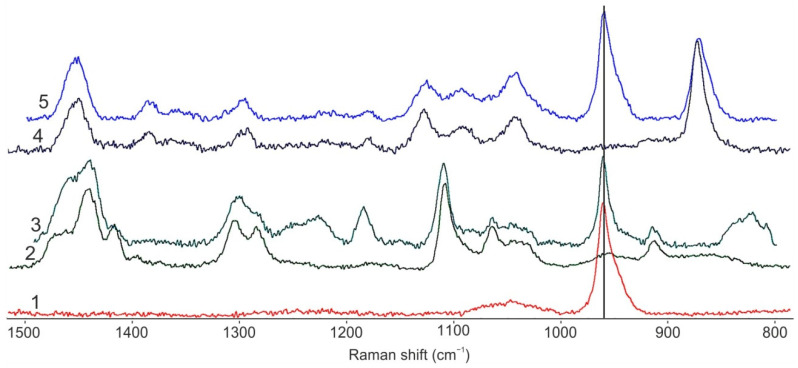
Raman spectra of initial nHA (1), PCL (2) and PCL-based microparticles contained nHA at polymer/nHA ratio of 1/1 (3); initial PLA (4) and PLA-based microparticles contained nHA at polymer/nHA ratio of 1/1 (5).

**Figure 11 polymers-14-04309-f011:**
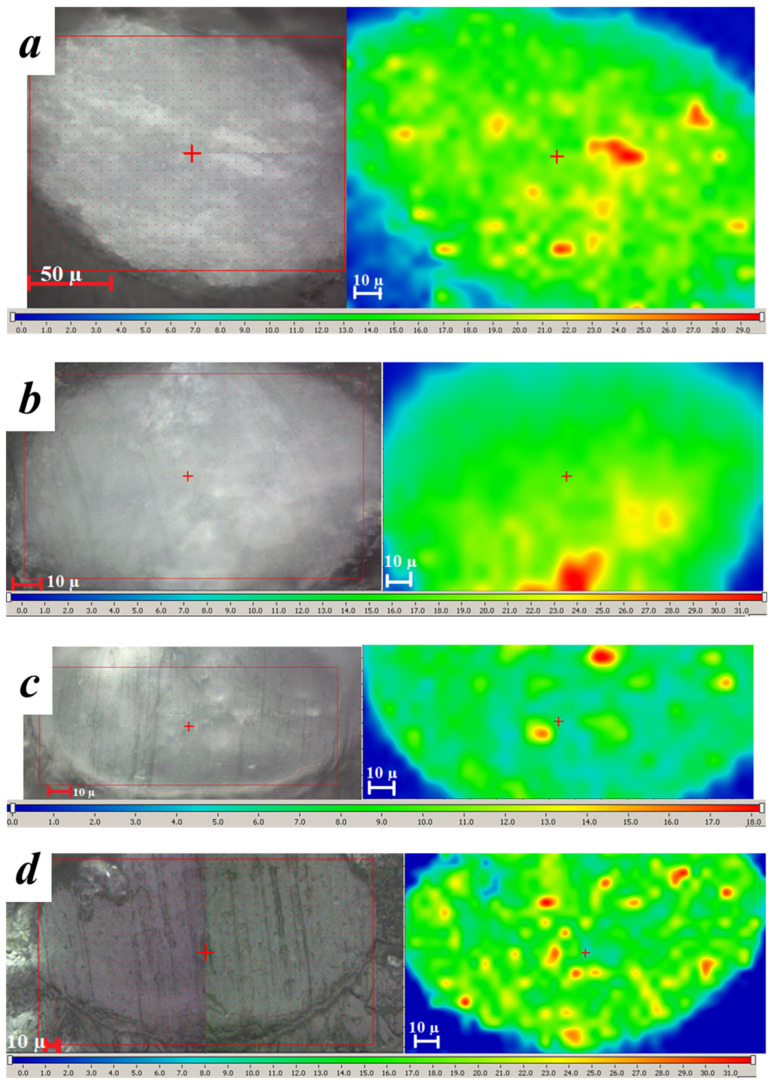
Images of the cross-section of the microparticles (on the left) and Raman maps of the nHA distribution within the studied microparticles (on the right). The red rectangle shows the location of the Raman map at the respective images. Microparticles based on (**a**,**b**) PLA or (**c**,**d**) PCL polymer solution containing polymer/nHA ratio as 1/1 (**a**,**c**) without preliminary US treatment or (**b**,**d**) with it.

**Figure 12 polymers-14-04309-f012:**
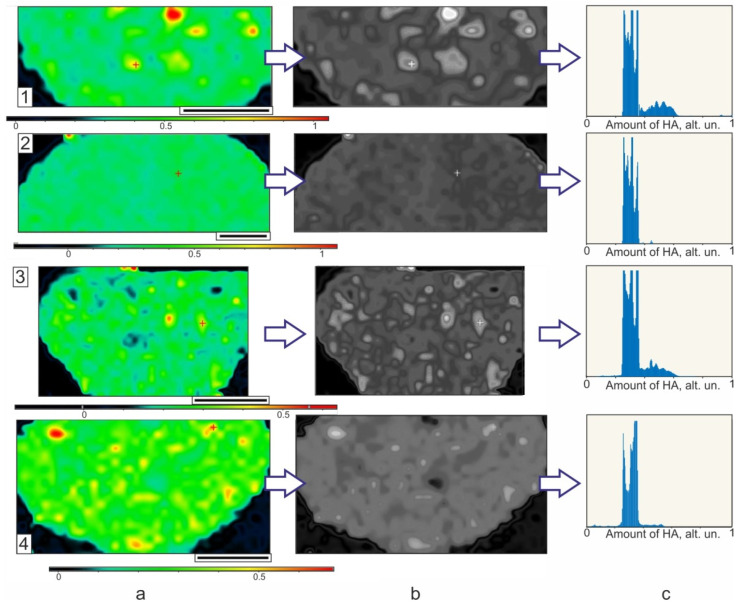
Raman maps (**a**) in color and (**b**) in grayscale form, and (**c**) calculated histograms of the nHA distribution within the volume of PCL microparticles made at polymer/nHA ratio as 1/1 without (1,2) and with (3,4) preliminary US treatment of oil phase.

## Data Availability

Data is contained within the article or Appendix A.

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
