# Peer review of "Controlled Structure of Polyester/Hydroxyapatite Microparticles Fabricated via Pickering Emulsion Approach"

_polymers, 2022, doi:10.3390/polym14204309_

Round 1
Reviewer 1 Report
Your article Preparation of Controlled structure of polyester/hydroxyapatite microparticles fabricated via Pickering emulsion approach. In my opinion, the Preparation of the polyester/hydroxyapatite composite process must be explained in detail. I hope readers will enjoy reading this article. I have some minor comments below. The authors are advised to take into consideration the following suggestions:
1) Authors should carefully revise and corrected all the grammatical issues and follow the scientific norms in the whole manuscript
2) Quality of figures can be improved, if possible
3) Please more elaborate on the novel aspect of your work at the end of the introduction.
4) Please use updated and recent papers in the literature review to give more sense to the reader.
5) Conclusions could be more specific and to the point, I would suggest looking and thinking about it.
6). Some of the introductions on this topic and … are need using from below papers, so, below papers are added to manuscript Recent publications
Recent progress in electrochemical detection of human papillomavirus (HPV) via graphene-based nanosensors
Bioactive Graphene Quantum Dots Based Polymer Composite for Biomedical Applications
Renewable Carbon Nano-materials: Novel Resources for Dental Tissue Engineering
Gold nanostars-diagnosis, bioimaging and biomedical applications
Data on cytotoxic and antibacterial activity of synthesized Fe3O4 nanoparticles using Malva sylvestris
3D Nanostructures for Tissue Engineering, Cancer Therapy, and Gene Delivery
Green Synthesis of Supermagnetic Fe3O4-MgO Nanoparticles via Nutmeg Essential Oil Toward Superior Anti-Bacterial and Anti-Fungal Performance
Asymmetric membranes: A potential scaffold for wound healing applications
Anti-bacterial/fungal and anti-cancer performance of green synthesized Ag nanoparticles using summer savory extract
Green Synthesis of Magnetic Nanoparticles Using Satureja hortensis Essential Oil toward Superior Antibacterial/Fungal and Anticancer Performance
Multifunctional gold nanorod for therapeutic applications and pharmaceutical delivery considering cellular metabolic responses, oxidative stress and cellular longevity
Recent advancements in polythiophene-based materials and their biomedical, geno sensor and DNA detection
Antibacterial effect of green synthesized silver nanoparticles using Ferula assafoetida against Acinetobacter baumannii isolated from the hospital and their cytotoxicity on the …
Bioactive agent-loaded electrospun nanofiber membranes for accelerating healing process: A review
Bio-Enhanced Polyrhodanine/Graphene Oxide/Fe3O4 Nanocomposite with Kombucha Solvent Supernatant as Ultra-Sensitive Biosensor for Detection of Doxorubicin Hydrochloride in
Enhancing the Physical, Mechanical, Oxygen Permeability and Photodegradation Properties of Styrene-acrylonitrile (SAN), Butadiene Rubber (BR) Composite by Silica Nanoparticles
Green Synthesis of Magnetic Nanoparticles Using Satureja hortensis Essential Oil toward Superior Antibacterial/Fungal and Anticancer Performance
Recent advancements in polythiophene-based materials and their biomedical, geno sensor and DNA detection
Author Response
Your article Preparation of Controlled structure of polyester/hydroxyapatite microparticles fabricated via Pickering emulsion approach. In my opinion, the Preparation of the polyester/hydroxyapatite composite process must be explained in detail. I hope readers will enjoy reading this article.
Answer: Thank you very much for the comment! We modified and extended the section on the microparticle fabrication process.
I have some minor comments below. The authors are advised to take into consideration the following suggestions:
1) Authors should carefully revise and corrected all the grammatical issues and follow the scientific norms in the whole manuscript
Answer: Thank you for your comment! We work on the manuscript to improve the English.
2) Quality of figures can be improved, if possible
Answer: We worked on the figures to improve their quality.
3) Please more elaborate on the novel aspect of your work at the end of the introduction.
Answer: Thank you for the suggestion. We highlighted the novelty within the abstract and at the end of the Introduction section.
4) Please use updated and recent papers in the literature review to give more sense to the reader.
Answer: Thank you for your comment! We carried additional literature search and added the literature.
5) Conclusions could be more specific and to the point, I would suggest looking and thinking about it.
Answer: We corrected the conclusions.
6). Some of the introductions on this topic and … are need using from below papers, so, below papers are added to manuscript Recent publications
Recent progress in electrochemical detection of human papillomavirus (HPV) via graphene-based nanosensors
Bioactive Graphene Quantum Dots Based Polymer Composite for Biomedical Applications
Renewable Carbon Nano-materials: Novel Resources for Dental Tissue Engineering
Gold nanostars-diagnosis, bioimaging and biomedical applications
Data on cytotoxic and antibacterial activity of synthesized Fe3O4 nanoparticles using Malva sylvestris
3D Nanostructures for Tissue Engineering, Cancer Therapy, and Gene Delivery
Green Synthesis of Supermagnetic Fe3O4-MgO Nanoparticles via Nutmeg Essential Oil Toward Superior Anti-Bacterial and Anti-Fungal Performance
Asymmetric membranes: A potential scaffold for wound healing applications
Anti-bacterial/fungal and anti-cancer performance of green synthesized Ag nanoparticles using summer savory extract
Green Synthesis of Magnetic Nanoparticles Using Satureja hortensis Essential Oil toward Superior Antibacterial/Fungal and Anticancer Performance
Multifunctional gold nanorod for therapeutic applications and pharmaceutical delivery considering cellular metabolic responses, oxidative stress and cellular longevity
Recent advancements in polythiophene-based materials and their biomedical, geno sensor and DNA detection
Antibacterial effect of green synthesized silver nanoparticles using Ferula assafoetida against Acinetobacter baumannii isolated from the hospital and their cytotoxicity on the …
Bioactive agent-loaded electrospun nanofiber membranes for accelerating healing process: A review
Bio-Enhanced Polyrhodanine/Graphene Oxide/Fe3O4 Nanocomposite with Kombucha Solvent Supernatant as Ultra-Sensitive Biosensor for Detection of Doxorubicin Hydrochloride in
Enhancing the Physical, Mechanical, Oxygen Permeability and Photodegradation Properties of Styrene-acrylonitrile (SAN), Butadiene Rubber (BR) Composite by Silica Nanoparticles
Green Synthesis of Magnetic Nanoparticles Using Satureja hortensis Essential Oil toward Superior Antibacterial/Fungal and Anticancer Performance
Recent advancements in polythiophene-based materials and their biomedical, geno sensor and DNA detection
Answer: Thank you for the suggestion. We updated the Introduction.
Reviewer 2 Report
Manuscript reference: polymers-1935446
Dear Editor,
The manuscript entitled: “Controlled structure of polyester/hydroxyapatite microparticles fabricated via Pickering emulsion approach” describes the development of microparticles with PLA or PCL and hydroxyapatite through Pickering emulsion production. The combination of hydroxyapatite with PLA and PCL has been studied to produce biocompatible materials; however, it is an interesting topic that should be studied since it enables biodegradable polyester/hydroxyapatite microparticles production for different biocompatible applications. The authors produce the microparticles and perform their characterization. The manuscript is well structured and reports an interesting study; however, the work can be improved considering the following observations:
1 – Revise the manuscript, mainly the abstract and conclusion sections, to improve the English.
2 – The introduction section is well, but the authors are encouraged to improve the novelty of the manuscript by describing better the main objectives of the work.
3 – The materials and methods section needs to be improved mainly the microparticles fabrication. What is the used oil phase – line 100, and was rapidly added – line 101, how? Its are important parameters to be clarified in the manuscript.
4 – Line 102 – the authors mention solvent evaporation, but how was this step carried out? What were the procedures? It should be clarified in the manuscript.
5 – Line 108 “The nHA particles were dispersed within water (mQ) using a magnetic stirrer at 200 rpm for 10 min. Additional ultrasound (US) treatment was used with an aim of immersion ultrasonic dispergator (23 kHz, 3 min) (I-10/0.63, LLC Ultrasonic technique – INLAB, Russia).” Was used two different steps for dispersing hydroxyapatite in water, why? What is the advantage of the ultrasound after stirring? It should be clarified in the manuscript.
6 – The authors mention biocompatibility issues and use CH2Cl2 and acetone as solvent. How much time does to evaporate, problems with toxicity, the solvents are reused? It should be clarified in the manuscript.
7 – Rewrite the following paragraph: Line 127 to 129.
8 - The authors show two production ways of the microparticles. But, it is not clear the main differences between them for non- and filled hydroxyapatite. Also, the authors are encouraged to explain better the main goal to produce filled microparticles.
9 – Section 2.3: “The evolution of Pickering emulsions from droplets to microparticles was monitored 142 under optical microscopic observation starting from 5 min after addition of oil phase into 143 aqueous phase and afterward every 10 min.” The authors should provide how much time is needed. ? It should be clarified in the manuscript.
10 – In the manuscript, hydroxyapatite appears in different forms: nHA, HA, and hydroxyapatite throughout the manuscript, it should be normalized.
11 – Paragraphs of lines 173 to 177 should be rewritten because the idea is not perceptible.
12 - Paragraphs of lines 213 to 216 should be rewritten. Why does hydroxyapatite become rigid when treated with ultrasound? Does the ultrasound not separate the possible aggregates of the hydroxyapatite? A better explanation must be added to the manuscript to clarify this topic.
13 – The authors are encouraged to explain why did the increase in HA lead to the formation of irregularly shaped microparticles? Line 269 – 271.
14 – Lines 275 – 277 - How do the authors confirm that dichloromethane and acetone are completely evaporated. Have the authors carried out any study of the relationship between the evasion rate and the shape of the morphology? Authors are encouraged to insert information in the manuscript in order to clarify the reader.
15 – In Figure 6, the authors indicate that hydroxyapatite particles are surrounded by the polymeric core; however, a similar type of deposition or similar appearance can be observed on the sides of the tube. The authors can clarify this point?
16 – The type of graphs in Figures 3 and 7 makes it difficult to analyze the data in the figure. Also, in Figure 3 why the total yield is higher than 120?
17 – Miscellaneous in line 326.
Author Response
The manuscript entitled: “Controlled structure of polyester/hydroxyapatite microparticles fabricated via Pickering emulsion approach” describes the development of microparticles with PLA or PCL and hydroxyapatite through Pickering emulsion production. The combination of hydroxyapatite with PLA and PCL has been studied to produce biocompatible materials; however, it is an interesting topic that should be studied since it enables biodegradable polyester/hydroxyapatite microparticles production for different biocompatible applications. The authors produce the microparticles and perform their characterization. The manuscript is well structured and reports an interesting study; however, the work can be improved considering the following observations:
1 – Revise the manuscript, mainly the abstract and conclusion sections, to improve the English.
Answer: Thank you for your comment! We work on the manuscript to improve the English.
2 – The introduction section is well, but the authors are encouraged to improve the novelty of the manuscript by describing better the main objectives of the work.
Answer: Thank you for the comment! We worked on the Introduction section. The main novelty of this work is fabrication of polyester-based microparticles filled with nHA via Pickering emulsion solvent evaporation technique.
3 – The materials and methods section needs to be improved mainly the microparticles fabrication. What is the used oil phase – line 100, and was rapidly added – line 101, how? Its are important parameters to be clarified in the manuscript.
Answer: In the case of core/shell microparticles the oil phase consisted of polymeric solution, while nHA was added within the aqueous phase. The microparticles bulk-filled with nHA were fabricated using dispersion of nHA in polymer solution as the oil phase. The oil phase was added within aqueous phase using autopipette. We modified this part of materials and methods section to make it clearer.
4 – Line 102 – the authors mention solvent evaporation, but how was this step carried out? What were the procedures? It should be clarified in the manuscript.
Answer: Thank you for the comment! This oil/water solvent evaporation technique is the classical method of PLA microparticles fabrication described in [Campos, E.; Branquinho, J.; Carreira, A.S.; Carvalho, A.; Coimbra, P.; Ferreira, P.; Gil, M.H. Designing Polymeric Microparticles for Biomedical and Industrial Applications. Eur. Polym. J. 2013, 49, 2005–2021, doi:10.1016/j.eurpolymj.2013.04.033.; O’Donnell, P.B.; McGinity, J.W. Preparation of Microspheres by the Solvent Evaporation Technique. Adv. Drug Deliv. Rev. 1997, 28, 25–42, doi:10.1016/S0169-409X(97)00049-5.; Blasi, P. Poly(Lactic Acid)/Poly(Lactic-Co-Glycolic Acid)-Based Microparticles: An Overview. J. Pharm. Investig. 2019, 49, 337–346, doi:10.1007/s40005-019-00453-z.]. The final microparticles are formed from oil/water emulsions as the solvent from the oil phase (CH2Cl2 + acetone) is transported out from the initial oil droplets through the aqueous phase and evaporated at the emulsion–air interface [Rosca, I.D.; Watari, F.; Uo, M. Microparticle Formation and Its Mechanism in Single and Double Emulsion Solvent Evaporation. J. Control. Release 2004, 99, 271–280, doi:10.1016/j.jconrel.2004.07.007.].
5 – Line 108 “The nHA particles were dispersed within water (mQ) using a magnetic stirrer at 200 rpm for 10 min. Additional ultrasound (US) treatment was used with an aim of immersion ultrasonic dispergator (23 kHz, 3 min) (I-10/0.63, LLC Ultrasonic technique – INLAB, Russia).” Was used two different steps for dispersing hydroxyapatite in water, why? What is the advantage of the ultrasound after stirring? It should be clarified in the manuscript.
Answer: Thank you for the question. Indeed, we used commercial hydroxyapatite nanoparticles with a size below 200 nm (Sigma-Aldrich, CAS: 12167-74-7, Lot #MKBP9910V), which were sold in a dry state. To disperse the nanoparticles in the phase we used either magnetic stirring only or additionally treated the nHA-containing phase by US ultrasound. This US treatment was aimed to disaggregate the nanoparticles. Dynamic laser scattering of the nanoparticles dispersed in water by magnetic stirring showed that the nanoparticles were in a form of aggregates with a mean size of 1.2 mm. Additional US treatment of this water dispersions allowed to decrease the size up to ~100 nm, i.e. to disperse the nanoparticles up to individual particles. We compared how the method of the nanoparticle dispersing (magnetic stirring or additional US treatment as well) affects the characteristics of the fabricated microparticles. We worked on the manuscript to clarify it.
6 – The authors mention biocompatibility issues and use CH2Cl2 and acetone as solvent. How much time does to evaporate, problems with toxicity, the solvents are reused? It should be clarified in the manuscript.
Answer: Dichloromethane (1.8 mL) and acetone (0.2 mL) evaporated for at least 3 hrs in a course of emulsion solidification. Then, the microparticles were washed with mQ water, sieved and dried. The microparticles fabricated by this technique using CH2Cl2 and acetone are biocompatible as it was shown previously in [10.3390/molecules25081949]. The total amount of the solvents in oil phase (CH2Cl2 + acetone) for one experiment was 2 mL, and we didn’t reuse it.
7 – Rewrite the following paragraph: Line 127 to 129.
Answer: The paragraph was rewritten.
8 - The authors show two production ways of the microparticles. But, it is not clear the main differences between them for non- and filled hydroxyapatite. Also, the authors are encouraged to explain better the main goal to produce filled microparticles.
Answer: Thank you for the comment! Two types of hybrid nHA-containing microparticles were fabricated: (1) core/shell microparticles with polymeric core and nHA shell; (2) polymeric microparticles filled with nHA over the whole volume. The main processing difference between core/shell and filled microparticles is the phase into which the nHA were added: water or oil phase, respectively. These hybrid microparticles could serve as micro-scaffolds (i.e., cell microcarriers) for bone tissue regeneration and simultaneous drug delivery. In the case of core/shell microparticles, the inorganic shell may serve for regulating a drug release profile [J. Mater. Sci. Mater. Med. doi:10.1007/S10856-011-4250-6; Mater. Sci. Eng. C. Mater. Biol. Appl. doi:10.1016/J.MSEC.2013.01.048] and improving cell adhesion to the microparticles surface [Langmuir, doi:10.1021/la901100z]. The nHA-filled microparticles are could be also used as a starting material for fabrication of larger 3D scaffolds for bone tissue engineering via molding or additive technologies, such as laser sintering. The distribution of nHA over the whole volume of the microparticles provides the constant supply of the mineral component during the polymeric scaffold degradation. We updated the manuscript to clarify this idea.
9 – Section 2.3: “The evolution of Pickering emulsions from droplets to microparticles was monitored 142 under optical microscopic observation starting from 5 min after addition of oil phase into 143 aqueous phase and afterward every 10 min.” The authors should provide how much time is needed. ? It should be clarified in the manuscript.
Answer: The total time of the microparticles fabrication is 3hrs. Normally, the droplets are visually solidified in approx. 2 hrs, but we are monitoring the systems until the end (3 hrs). This information was added within the revised version of the manuscript.
10 – In the manuscript, hydroxyapatite appears in different forms: nHA, HA, and hydroxyapatite throughout the manuscript, it should be normalized.
Answer: Thank you for the comment! We try to normalize it throughout the manuscript. In the Introduction section we also mention the works on fabrication of hydroxyapatite-containing microparticles, where hydroxyapatite was precipitated onto microparticles surface using biomineralization approach. In such cases we had to use HA.
11 – Paragraphs of lines 173 to 177 should be rewritten because the idea is not perceptible.
Answer: Thank you! The paragraph was rewritten.
12 - Paragraphs of lines 213 to 216 should be rewritten. Why does hydroxyapatite become rigid when treated with ultrasound? Does the ultrasound not separate the possible aggregates of the hydroxyapatite? A better explanation must be added to the manuscript to clarify this topic.
Answer: These paragraphs were rewritten. Ultrasound separates the nHA aggregates up to individual particles, which are adsorbed at the oil/water interface. During the process of solvent evaporation the interface area is decreased, but the rearrangement of the interface is limited due to adsorbed nHA layer. The same morphology was shown in [Tham, C.Y.; Chow, W.S. Poly(Lactic Acid) Microparticles with Controllable Morphology by Hydroxyapatite Stabilized Pickering Emulsions: Effect of PH, Salt, and Amphiphilic Agents. Colloids Surfaces A Physicochem. Eng. Asp. 2017, 533, 275–285]. In contrast to their work, we also studied the PLA stabilization with nHA aggregates, i.e., without preliminary US treatment of the aqueous phase. In this case, the oil/water interface could be rearranged by splitting up the nHA aggregates. We added the Figure within the revised version of the manuscript to illustrate the proposed mechanism of the interface rearrangement.
13 – The authors are encouraged to explain why did the increase in HA lead to the formation of irregularly shaped microparticles? Line 269 – 271.
Answer: Thank you for the comment! We suppose that in the case of insertion of nHA within oil polymeric phase, the nHA work as framework, which is limiting the oil droplets reorganization during solvent evaporation.
14 – Lines 275 – 277 - How do the authors confirm that dichloromethane and acetone are completely evaporated. Have the authors carried out any study of the relationship between the evasion rate and the shape of the morphology? Authors are encouraged to insert information in the manuscript in order to clarify the reader.
Answer: The microparticles fabricated by this technique using CH2Cl2 and acetone are biocompatible [T.S. Demina, M.G. Drozdova, C. Sevrin, P. Compère, T.A. Akopova, E. Markvicheva, C. Grandfils Biodegradable Cell Microcarriers Based on Chitosan/Polyester Graft-Copolymers // Molecules 2020, 25, 1949; doi:10.3390/molecules25081949], so we suppose that the dichloromethane and acetone were evaporated. The solvents (2 mL) were evaporated for 3 hrs during emulsion fabrication; the samples were washed with mQ and freeze-dried for approx. 3 days. We added this information within the revised version of the manuscript. Regarding to the relationship between the evasion (evaporation?) rate and the shape of the morphology: we didn’t study it especially, but such type of relationship is described in literature [Yang Y, Fang Z, Chen X, Zhang W, Xie Y, Chen Y, Liu Z and Yuan W (2017) An Overview of Pickering Emulsions: Solid-Particle Materials, Classification, Morphology, and Applications. Front. Pharmacol. 8:287. doi: 10.3389/fphar.2017.00287].
15 – In Figure 6, the authors indicate that hydroxyapatite particles are surrounded by the polymeric core; however, a similar type of deposition or similar appearance can be observed on the sides of the tube. The authors can clarify this point?
Answer: We confirmed the presence of nHA shell at the surface of the microparticles by energy-dispersive X-ray spectroscopy carried out on the whole microparticles (Figure S1 in Supplementary Material). The Figure 6 (within the revised version is Figure 7) shows the evaluation of polymeric core composition by Raman spectroscopy. The appearance of microparticle surface at its cross-section and the sides of the tube is similar due to SEM features.
16 – The type of graphs in Figures 3 and 7 makes it difficult to analyze the data in the figure. Also, in Figure 3 why the total yield is higher than 120?
Answer: We agree that the Figure 3 and 7 are a little bit overloaded with data. We try to represent the total yield and size distribution of the microparticles within one type of graph. We modified the figures to improve their quality. Indeed, the total yield could by higher than 100% due to entrapment of nHA at the oil/water interface and, therefore, at the surface of microparticles after evaporation of the solvents from the oil phase. 100 wt.% is a total weight of components dissolved within the oil phase, i.e. PLA or PCL (in the case of core/shell microparticles).
17 – Miscellaneous in line 326.
Answer: Corrected.
We would like to thank the Referee for his/her time and concern in the analysis of our manuscript and to the valuable advices received.
Round 2
Reviewer 2 Report
The authors tried to answer all questions.
The Manuscript has been improved.
Author Response
Thank you very much for your comments and suggestions that allowed us to improve the manuscript.